# Green Route for the Isolation and Purification of Hyrdoxytyrosol, Tyrosol, Oleacein and Oleocanthal from Extra Virgin Olive Oil

**DOI:** 10.3390/molecules25163654

**Published:** 2020-08-11

**Authors:** Antonio Francioso, Rodolfo Federico, Anna Maggiore, Mario Fontana, Alberto Boffi, Maria D’Erme, Luciana Mosca

**Affiliations:** 1Department of Biochemical Sciences, “Sapienza” University of Rome, Piazzale Aldo Moro 5, 00185 Rome, Italy; anna.maggiore@uniroma1.it (A.M.); mario.fontana@uniroma1.it (M.F.); alberto.boffi@uniroma1.it (A.B.); maria.derme@uniroma1.it (M.D.); luciana.mosca@uniroma1.it (L.M.); 2MOLIROM s.r.l, via Carlo Bartolomeo Piazza 8, 00161 Rome, Italy; r.federico@molirom.com

**Keywords:** extra virgin olive oil, oleocanthal, secoiridoids, green chemistry, hydoxytyrosol, tyrosol, oleacein, natural deep eutectic solvents, ultra-high-performance liquid chromatography, mass spectrometry

## Abstract

Extra virgin olive oil (EVOO) phenols represent a significant part of the intake of antioxidants and bioactive compounds in the Mediterranean diet. In particular, hydroxytyrosol (HTyr), tyrosol (Tyr), and the secoiridoids oleacein and oleocanthal play central roles as anti-inflammatory, neuro-protective and anti-cancer agents. These compounds cannot be easily obtained via chemical synthesis, and their isolation and purification from EVOO is cumbersome. Indeed, both processes involve the use of large volumes of organic solvents, hazardous reagents and several chromatographic steps. In this work we propose a novel optimized procedure for the green extraction, isolation and purification of HTyr, Tyr, oleacein and oleocanthal directly from EVOO, by using a Natural Deep Eutectic Solvent (NaDES) as an extracting phase, coupled with preparative high-performance liquid chromatography. This purification method allows the total recovery of the four components as single pure compounds directly from EVOO, in a rapid, economic and ecologically sustainable way, which utilizes biocompatible reagents and strongly limits the use or generation of hazardous substances.

## 1. Introduction

Extra virgin olive oil (EVOO) represents one of the most important sources of biologically active compounds and antioxidants in the Mediterranean diet [1,2,3,4,5]. EVOO polyphenols and secoiridoids possess important biological effects, and are extremely interesting and promising compounds from a pharmacological point of view [3,6,7,8]. Hydroxytyrosol (HTyr), tyrosol (Tyr), oleacein and oleocanthal are the main compounds responsible for the beneficial effects of EVOO as part of the Mediterranean diet (Figure 1) [9,10].

These compounds possess a strong antioxidant activity, and HTyr and oleocanthal in particular have a synergistic marked anti-inflammatory effect [11,12,13,14]. Tyr instead possesses important central nervous system and cardioprotective effects [9,15,16,17,18,19,20]. Oleocanthal and its less studied cathecolic analog, oleacein, are nowadays two of the most interesting bioactive natural products under biological investigation [10]. Oleocanthal is responsible for the particular spicy organoleptic sensation that EVOO induces on the tongue. The scientific relevance of and interest in oleocanthal is mainly due to its potent anti-inflammatory effect (several times more active than other FANS, e.g., Ibuprofen), its selective cytotoxicity for cancer cells, and the very promising neuroprotective activity that oleocanthal has demonstrated in neurodegenerative pathologies, i.e., Alzheimer’s disease [6,7,13,14,21,22,23,24,25,26,27]. Given all these considerations, these compounds are extremely interesting and promising for the preparation of dietary supplements, nutraceuticals, functional foods or cosmeceuticals.

Despite their promising pharmacological activities, there is still a limited number of studies that have been carried out on oleocanthal and oleacein, as these molecules are not easily available on the market, and even when available are extremely expensive. The difficulty of obtaining large amounts of pure compounds at affordable prices is principally due to their expensive and difficult purification from EVOO. Oleocanthal and oleacein are not present in *Olea europaea* L. plants (leaves and fruits), but are formed during EVOO production and processing via the conversions of oleuropein and ligstroside, two glycosidic secoiridoids derived from HTyr and Tyr, respectively (Figure 2) [16,28,29,30,31,32].

The isolation of oleocanthal and oleacein requires the extensive treatment of EVOO with organic solvents such as *n*-hexane and acetonitrile, and subsequently several chromatographic separation steps (mostly silica based) and the use of organic eluents such as dichloromethane, petroleum ether and ethyl acetate [33,34,35,36]. Even by using large volumes of organic solvents in the extraction, the gravity columns and the preparative thin layer chromatography, the yields and the purity of the products are often low, expensive, and moreover not ecologically friendly and biocompatible, and do not eliminate the use or generation of hazardous substances. With this work, we propose a novel strategy for the green extraction and isolation of HTyr, Tyr, oleacein and oleocanthal in a single chromatographic step, using natural deep eutectic solvent (NaDES) extraction coupled with preparative high-performance liquid chromatography. Our method allows an efficient and complete extraction of the aforementioned compounds, obtaining highly concentrated, high final yields of the pure products with the use of biocompatible reagents, leaving no hazardous organic residues after the purification.

## 2. Results

### 2.1. Phenols and Secoiridoids Characterization in EVOO/NaDES Extract

Our results show that EVOO treatment with NaDES (Choline:Glycerol, 1:1.5 molar ratio) allowed us to obtain highly concentrated solutions of HTyr, Tyr, oleacein and oleocanthal by extracting them directly with a 1:20 ratio (*w*/*w*) of NaDES:EVOO. The UPLC-PDA/MS analysis of EVOO NaDES extracts revealed the presence of HTyr, Tyr, oleacein and oleocanthal as major components, together with other minor products eluted in the last part of the hydrophobic chromatographic gradient (Figure 3). Compounds were identified by their retention time, UV-visible and MS spectra, and by comparison with pure analytical standards.

As shown by the UPLC chromatogram (Figure 3), the first two eluted compounds were identified as HTyr and Tyr, with retention times of 2.8 and 3.6 min, respectively. The identity of these compounds was verified by the analysis of their UV-visible spectra, and by the injection of the analytical standards. The MS spectra under our conditions were not available due to the lower ionization features of these two phenols, with respect to their secoiridoids derivatives. The catechol moiety of HTyr has a characteristic UV-visible spectrum, with a λ_max_ at 280 nm, whereas the mono substituted phenol Tyr displays a UV-visible λ_max_ at a lower wavelength (275 nm), with a shoulder at 280 nm (Figure 4).

The oleacein and oleocanthal in NaDES extracts were recognized firstly by analysis of their UV-visible and MS spectra, which was then confirmed by analytical standard injection. These two phenolic secoiridoids possess the same chromophoric moiety as their precursors (catechol for HTyr and phenol for Tyr), with the same UV-visible spectra and different characteristic MS negative ionization spectra (Figure 5). Oleacein elutes at 5.9 min, and its pseudo-molecular ion displays an [M−H]^−^ peak at 319.08 *m*/*z*. Oleocanthal (retention time, 6.4 min) possess the same UV-visible spectrum as Tyr, with a pseudomolecular ion [M−H]^−^ at 303.08 *m*/*z* in the same negative ESI-MS mode.

Though not of interest in this work, another minor compound eluted at 6.9 min (Figure 2) was detected, and tentatively characterized by means of chromatographic retention factor, UV-visible spectrum and MS spectrum. The compound displays the UV-visible features of EVOO catechols, and an [M−H]^−^ pseudomolecular ion at 377.11 *m*/*z*, as reported in the literature for oleuropein aglycone [37,38,39](Figure 5).

### 2.2. Extraction, Determination and Compounds Isolation

EVOO was extracted three times with NaDES, and after each extraction step the compounds of interest were determined and quantified in the extracts.

The UPLC chromatographic profiles and calibration curves of the HTyr, Tyr, oleacein and oleocanthal analytical standards are shown in Figure 6.

After the first extraction, EVOO was re-extracted two more times with the same ratio of NaDES:EVOO (1:20 *w*/*w*) to evaluate the efficiency of the extraction and the relative recovery of the single compounds after each step. Our results show that the first extraction allows the recovery of almost all of the four compounds, with almost 100% yield for the catechol compounds (HTyr and oleacein; 100% and 97%, respectively) and a slightly lower yield for Tyr and oleocanthal (90% and 70%, respectively). The second extraction recovers the remaining amount of all the compounds, and the last one allows the complete recovery of the last residues of oleocanthal (Figure 7).

The first extract was purified via reverse phase preparative HPLC. The second and the third extracts were combined together and purified by the same process. Before injection, each extract was added to a 1.5 volume of ultrapure water (1:1.5 extract:water) to permit feasible injection with no significant increase of backpressure caused by the high viscosity of NaDES compared to pure water. The elution was performed with an ethanol gradient in ultrapure water and, as shown in Figure 8, the peak selectivity and chromatographic behavior were maintained at the same levels as those of the UPLC analytical reverse phase system.

The peaks of interest were collected and the solvent was reduced by rotary evaporation. Dry pure compounds were then recovered from aqueous elutes by freeze-drying. After the overall purification process, from 1 kg of EVOO we obtained ca. 102 mg of oleocanthal, 82 mg of oleacein, 11 mg of Tyr and 9 mg of HTyr.

## 3. Discussion

With this work we developed and set up a novel method for the purification of HTyr, Tyr, oleacein and oleocanthal from EVOO, using a completely green and simple two step procedure. Previous papers reported the extraction of EVOO polyphenols using deep eutectic solvents for the analytical determination of EVOO polyphenols, but no studies were carried out on the application and optimization of the extraction and preparative purification of these four compounds [40,41,42,43,44]. NaDES extraction allowed us to obtain highly concentrated green solutions of HTyr, Tyr, oleacein and oleocanthal, which were identified and determined by analytical standards authentication and by means of UPLC-DAD/MS. The chromatographic analysis evidenced that catechol compounds (HTyr and oleacein) both elute before their respective mono substituted phenolic analogues (Tyr and oleocanthal), due to their more hydrophilic nature and the minor retention factors in reverse stationary phases. The actual method involves three subsequent extractions of EVOO, which afford a total recovery of the molecules, with different relative amounts of each compound at each step. The first extraction leads to the recovery of almost all the simple phenols (HTyr and Tyr) and oleacein, and about 70% of the oleocanthal. The remaining part of oleocanthal is purified by pooling together the second and the third extractions, in which the compound is derived as the major constituent. The strategy of using very concentrated NaDES phenolics and secoirodoids extracts gives several advantages. The first one is the obtainment of highly concentrated preparations in a non-volatile medium that can be further easily processed. The second advantage is the different relative extraction power of NaDES in the given ratio with EVOO (1:20 *w*/*w*), which allows an enrichment of the first extract that improves the subsequent chromatographic separation steps. The third and the most interesting feature of this methodology is the possibility of using reverse phase preparative HPLC for the purification of such concentrated compounds in solution. Phenolics and secoiridoids are not such hydrophilic compounds, and the injection phase of these molecules into preparative RP-HPLC is often a problem. The use of NaDES aqueous extracts (1:1.5 extract:water)in the injection phase maintains the compounds in the hydrophilic phase, with no significant influence on solubility and injection, and no backpressure problems caused by the high viscosity of pure NaDES with respect to the initial mobile phase (water). With this method it is possible to purify ca. 200 mg of oleocanthal and oleacein from 1 kg of EVOO, with few RP-preparative HPLC chromatographic purifications and with a total recovery of the four compounds in a completely green process; a yield that, so far, is at least two times higher than that achieved by other reported methods [33,45]. As regards the phenolics precursors HTyr and Tyr, their amounts and concentrations in different EVOO depends not only on the *Olea Europea* L. cultivar, but also on the aging of the EVOO. It is even possible with this method to obtain higher yields of HTyr and Tyr by applying the purification process to aged EVOOs, in which the secoiridoids are converted back into their precursors over time. Our method finally gives the opportunity to many research laboratories to purify such scientifically interesting and very expensive compounds, via a completely green and efficient method, allowing the possibility of further studies in the biological, chemical and biochemical field.

## 4. Materials and Methods

### 4.1. Reagents and Chemicals

All the reagents and chemicals were purchased, at analytical grade, from Sigma-Aldrich (Milan, Italy). HPLC and LC-MS grade solvents used were purchased from Carlo Erba (Milan, Italy). HTyr, Tyr, oleacein and oleocanthal analytical standards were purchased from Sigma-Merck (Darmstadt, Germany). EVOO (*Olea Europea* L., Coratina cultivar) was obtained from Puglia region (Italy).

### 4.2. UPLC-DAD/MS

UPLC-DAD/MS was performed on a Waters Acquity H-Class UPLC system (Waters, Milford, MA, USA), including a quaternary solvent manager (QSM), a sample manager with a flow through needle system (FTN), a photodiode array detector (PDA) and a single-quadruple mass detector with electrospray ionization source (ACQUITY QDa). Chromatography was performed on a Phenomenex Kinetex C18 column (100 mm × 2.1 mm i.d., 2.6 μm particle size). Solvent A was 0.1% aqueous HCOOH and solvent B was 0.1% HCOOH in MeOH. The flow rate was 0.5 mL/min and column temperature was set at 35 °C. Elution was performed isocratically for the first minute with 2% B; from minute 1 to minute 6 solvent B was linearly increased to 55%; from 6 to 10 min 20% A and 80% B; then, for 0.5 min solvent B was set at 100% and maintained for 2 min. The column was re-equilibrated with 98% A and 2% B before the next injection. Samples were diluted in the mobile phase and injected through the needle. The PDA detector was set up in the range 200 to 600 nm. Mass spectrometric detection was performed in the negative electrospray ionization mode, using nitrogen as the nebulizer gas. Analyses were performed in the Total Ion Current (TIC) mode with a mass range of 50–1000 *m*/*z*. The capillary voltage was 0.8 kV, cone voltage 15 V, ion source temperature 120 °C and probe temperature 600 °C. Quantification of each compound was performed by using standards calibration curves in the range of 0.1–2 nmol in the column.

### 4.3. EVOO Extraction and Compounds Isolation

NaDES choline:glycerol (1:1.5 molar ratio) was prepared by mixing the two components at 60 °C under magnetic stirring for 1 h. After cooling EVOO was added to the NaDES in the ratio 1:20 *w*/*w* (NaDES:EVOO). The extractions were carried out under magnetic stirring at room temperature for 15 min and then transferred via a separatory funnel for decantation and phases separation. The same procedure was repeated 3 times to evaluate the recovery of single compounds after each extraction. After UPLC quantification of the phenolic and secoridoid components, the second and the third extractions were pooled together and these new extracts were each added in a ratio of 1.5 to bidistilled water. These two NaDES aqueous extracts were subsequently purified by preparative HPLC. Preparative chromatography was performed on a Waters HPLC 600 pump equipped with a controller and an UV-Vis photodiode array detector mod. 2996. The chromatographic column was a reverse phase Vydac C18 column, 22 mm × 250 mm, 10 μm particle size. Data analysis was monitored using a dedicated application (Millennium^32^). Elution was performed at 8 mL/min with a binary gradient system with bidistilled water (solvent A) and 80% aqueous ethanol (solvent B). The gradient was: 0–5 min, 2% B; 5–35 min, 100% B. During the separation the products were monitored spectrophotometrically at 280 nm and directly collected from the detector. After compound collection the solvents were evaporated under reduced pressure using a rotary evaporator, and finally dried by freeze-drying.

## 5. Conclusions

This work demonstrates the possibility of using a convenient, rapid and totally green novel methodology for the purification of HTyr, Tyr, oleacein and oleocanthal from EVOO by direct NaDES extraction, obtaining highly concentrated solutions of phenols and secoiridoids. These solutions possess several advantages with respect to classical organic extracts, especially for reverse phase preparative HPLC applications. The hydrophilic nature of NaDES allows its injection at large volumes into the starting reverse phase gradients, with no influence on the resolution and selectivity of the chromatography. As mentioned above, the present method, with respect to the currently available procedures, represents a completely green alternative for the isolation of EVOO phenolics and secoiridoids, which provides the total recovery of the compounds and prevents degradation processes caused by extensive organic extractions and different separation steps. Moreover, besides being an eco-friendly process, the proposed purification is even economically sustainable compared to traditional methods, in which large amounts of organic solvents are utilized and low yields and purities of the compounds are obtained. Due to the high costs and the pharmacological potential that EVOO phenols, and especially secoiridoids such as oleocanthal, display, the present method will represent a new solution for scientists who want to further investigate the chemical and biological properties of these natural products, of which there is still much to explore.

## Figures and Tables

**Figure 1 molecules-25-03654-f001:**
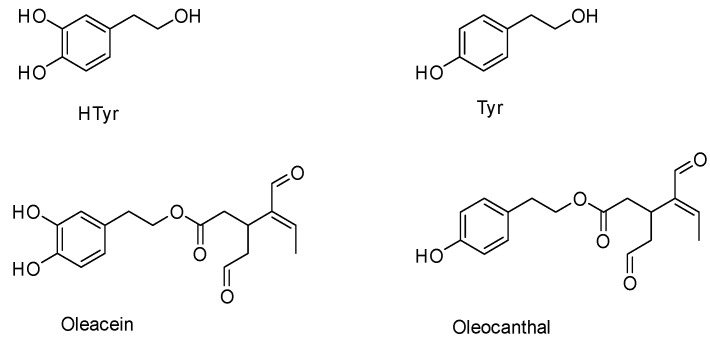
Chemical structure of EVOO secoiridoids and phenols.

**Figure 2 molecules-25-03654-f002:**
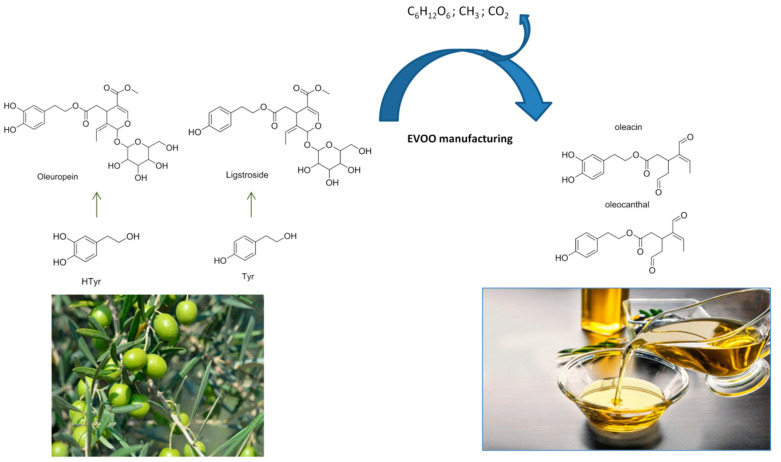
Oleocanthal and oleacein formation. During EVOO manufacturing, secoiridoids (oleuropein and ligstroside) are deglycosylated, and after demethylation and spontaneous decarboxylation free dialdehydes are produced.

**Figure 3 molecules-25-03654-f003:**
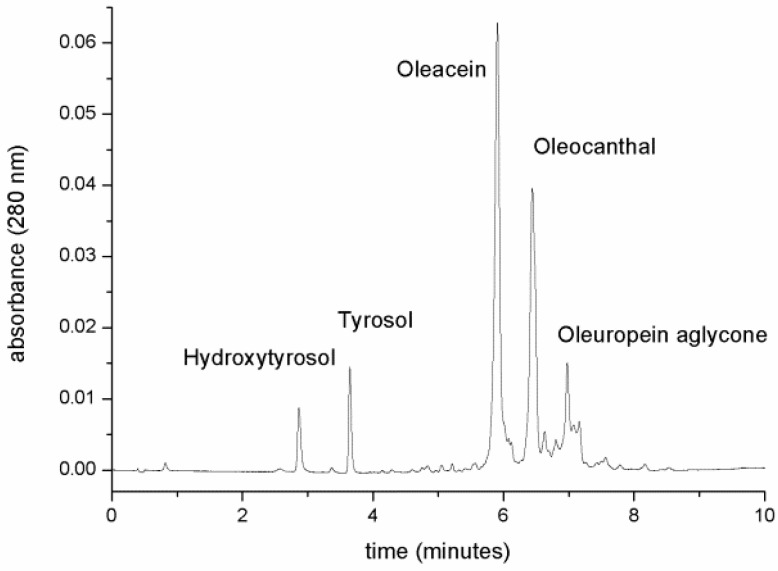
UPLC chromatographic analysis of EVOO/NaDES extract.

**Figure 4 molecules-25-03654-f004:**
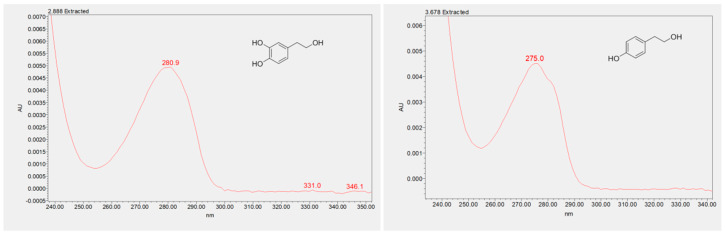
HTyr and Tyr UV-visible spectra.

**Figure 5 molecules-25-03654-f005:**
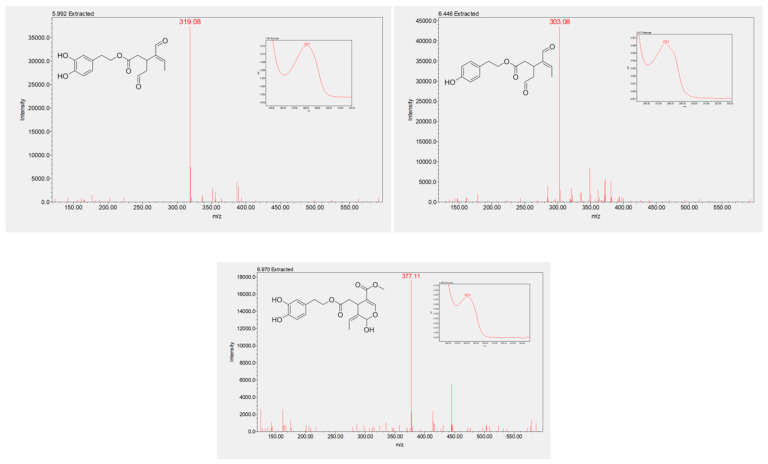
Oleacein (**left**), oleocanthal (**right**) and oleuropein aglycone (**down**) MS^−^ and UV-visible spectra (inset).

**Figure 6 molecules-25-03654-f006:**
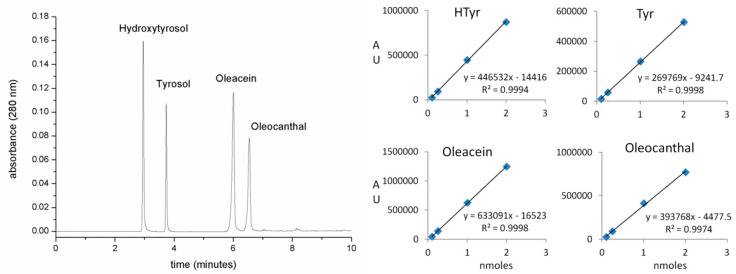
UPLC representative chromatogram (**left**) and calibration curves (**right**) of phenols and secoiridoids analytical standards.

**Figure 7 molecules-25-03654-f007:**
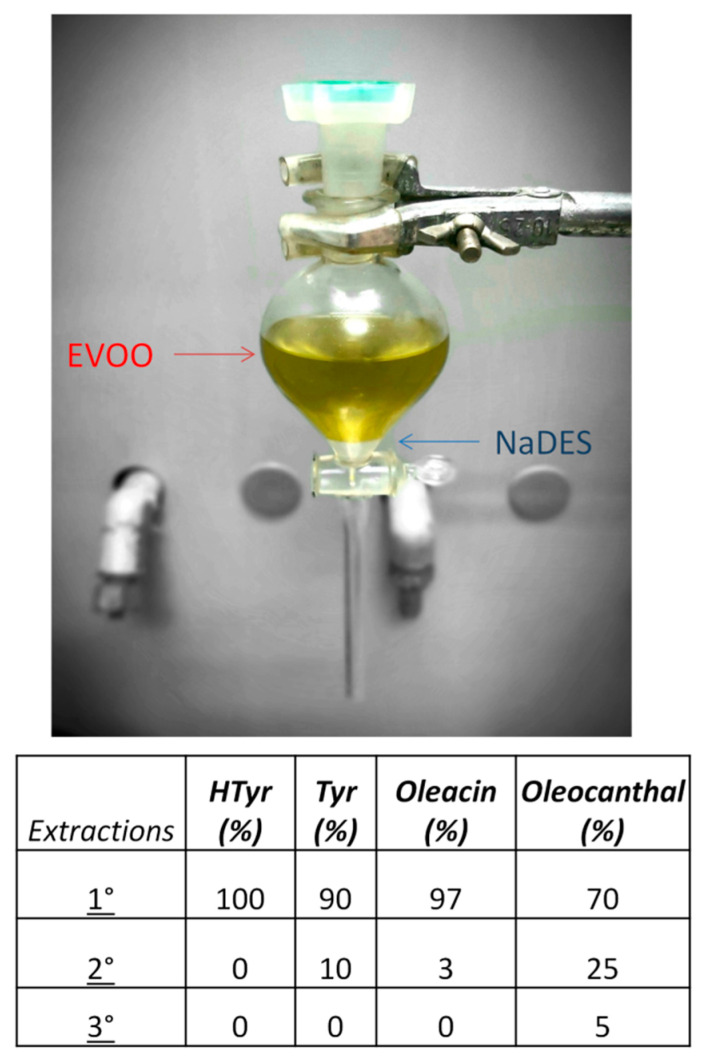
EVOO extraction with NaDES (**up**) and percentage recovery of the compounds after each extraction step (**bottom**)**.**

**Figure 8 molecules-25-03654-f008:**
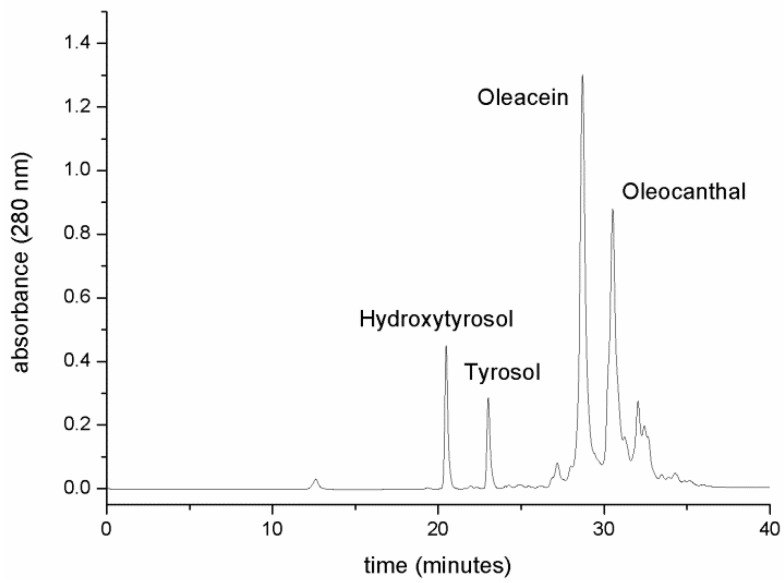
Reverse phase preparative HPLC purification of the compounds from NaDES extracts.

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
