# Peer review of "Green Route for the Isolation and Purification of Hyrdoxytyrosol, Tyrosol, Oleacein and Oleocanthal from Extra Virgin Olive Oil"

_molecules, 2020, doi:10.3390/molecules25163654_

Round 1
Reviewer 1 Report
The paper entitled " Green route for the Isolation and Purification of
Hyrdoxytyrosol, Tyrosol, Oleacein and Oleocanthal
from Extra Virgin Olive Oil"discussing a novel optimized procedure for the
green extraction, isolation and purification of HTyr, Tyr, oleacein and oleocanthal directly from EVOO by using a Natural Deep Eutectic Solvent (NaDES) as an extracting phase, coupled with preparative high performance liquid chromatography. This purification method allows the total recovery of the four components as single pure compounds directly from EVOO in a rapid, economic and ecologically sustainable way, that utilizes biocompatible reagents and strongly limits the use or generation of hazardous substances.
This manuscript is suffering major problem, where The main scientific concept addressed in this manuscript has been already reported in several publications that holding it from published anywhere, unless the author can show in details the uniqueness of this work compared to the others such as:
Bonacci S, Di Gioia ML, Costanzo P, Maiuolo L, Tallarico S, Nardi M. Natural Deep Eutectic Solvent as Extraction Media for the Main Phenolic Compounds from Olive Oil Processing Wastes. Antioxidants. 2020 Jun;9(6):513.
Author Response
We thank the referee for his comments. However, we would like to point out that the indicated work (Bonacci et al 2020) has been already cited by us as well as many others in which NaDES are used as an extraction solvents. Our paper is indeed based on the previous experiences of NaDES extraction and qualitative and quantitative analysis of different EVOOs. However, the purpose and evolution of our work, is precisely focused on the use of these solvents to isolate and purify in a preparative scale EVOO phenolics, and is not based on the development of an analytical method. As explained in the manuscript (lines 155-169 and Conclusions) the presented method represents the first totally green alternative to obtain the described pure compounds directly from EVOO with high yields compared to the previously described works and traditional organic chemistry purifications. The main advantage of the scaled preparative process is certainly the possibility to combine the NaDES extraction with the use of preparative HPLC. NaDES aqueous solutions can be directly injected into RP-preparative HPLC, offering the possibility of maintaining the molecules in solution during all the injection phase. One of the technical limits of the purification of natural products via preparative HPLC is often represented by HPLC injection solvents which must allow the complete solubilization of lipophilic compounds but at the same time must not affect the chromatographic separation. In this case the scaled up process using NaDES injection phase maintain the chromatographic resolution and allows a complete recovery of the compounds. The overall process presented in this work represents a valid alternative to overcoming several limits of EVOO phenolics purification in a totally green, biocompatible and eco-sustainable route.
Reviewer 2 Report
General comment: The research article entitled “Green route for the Isolation and Purification of Hyrdoxytyrosol, Tyrosol, Oleacein and Oleocanthalfrom Extra Virgin Olive Oil” is a well-organized study, with sufficient presentation and discussion of the issue. Some minor corrections are required for the improvement of the manuscript.
Abstract: The Abstract adequately presents the main objective of the study.
Introduction: The introduction section covers the majority of aspects relative to the need for new green methods for olive oil bioactives obtain.
Methodology: Authors describe adequately the methodology of the study.
Results and Discussion: The results are presented and discussed sufficiently.
-Could authors divide the discussion session into paragraphs?
-Could authors define possible limitations of the study?
Conclusion: The conclusion is adequate and summarizes the main text.
Bibliography/References: The references used by the authors cover adequately the relative scientific field and the aims of the study.
Author Response
Dear reviewer,
Thank you so much for your suggestions and for improving the quality of our manuscript.
As regards the limitations of presented methodology we want to point out that the main clue could be represented by the use of the HPLC equipment. All the preparation and purification steps are easily feasible and applicable in other laboratories, but as mentioned above an HPLC instrumentation is needed for the sepration of the compound. Apart from this important need the work describes the overall process and represents a novel way for the green purification of the four interesting EVOO bioactive natural products.
Thanks again for your comments
Best regards